



# The wind farm pressure field

Ronald B. Smith

Earth and Planetary Science Department, Yale University, New Haven, CT, 06520, USA

*Correspondence to*: Ronald B. Smith  (Ronald.smith@yale.edu)

**Abstract** The disturbed atmospheric pressure near a wind farm arises from the turbine drag forces in combination with vertical confinement associated with atmospheric stability.  These pressure gradients slow the wind upstream, deflect the air laterally, weaken the flow deceleration over the farm and modify the wake recovery.  Here, we describe the airflow and pressure disturbance near a wind farm under typical stability conditions and alternatively, with the simplifying assumption of a rigid lid. The rigid lid case clarifies the cause of the pressure disturbance and its close relationship to wind farm drag.

The key to understanding the rigid lid model is the proof that the pressure field p(x,y) is a Harmonic Function almost everywhere. It follows that the maximum and minimum pressure occur at the front and back edge of the farm. Over the farm, the favorable pressure gradient is constant and significantly offsets the turbine drag. Upwind and downwind of the farm, the pressure field is a dipole given by $p(x,y) \approx Axr^{-2}$ where the coefficient $A$  is proportional to the total farm drag. Two derivations of this law are given. Field measurements of pressure can be used to find the coefficient A and thus to estimate

total farm drag.

## 1.    Introduction

The construction of offshore wind farms may significantly help our society transition to renewable energy but the wind slowing

by these farms may ultimately limit their potential for electric power generation (Ahktar et al. 2022). This issue has an extensive literature, reviewed recently by Stevens and Meneveau (2017), Archer et al. (2018), Porte-Agel et al. (2020), Pryor et al. (2020), Fischereit et al. (2021). An integral part of the wind slowing by turbine drag is the creation of a local pressure field. This pressure disturbance was initially neglected (Jensen 1983) but has been recently estimated in connection with gravity wave (GW) generation (Smith 2010, 2022, Wu and Porté-Agel, 2017, Allaerts and Meyers, 2018, 2019). In a stably stratified

atmosphere, the lifting of the air caused by farm drag creates gravity waves aloft whose pressure field acts back on the lower atmosphere.

This pressure field modifies the airflow in ways that the direct action of turbine drag cannot. First, it can decelerate the flow before it reaches the first row of turbines, so call "Blocking" (Bleeg et al. 2018).  Second, it can deflect the air to the left and right. Third, over the farm, it can fight back against the turbine drag, helping to keep the wind flow strong. Finally,  it alters

the recovery of the wake.

The pressure field near a wind farm is analogous in some respects to that for a single turbine. The subsonic airflow approaching a turbine disk begins to decelerate upwind due to an adverse pressure gradient and its corresponding  "axial induction factor"



reduces the turbine efficiency to the Betz limit (Hanson, 2000). According to Gribben and Hawkes (2019), the local non-hydrostatic pressure disturbances decays inversely as the square of the distance upstream. The farm-generated hydrostatic
pressure disturbance is more far-reaching.

In this paper, we compare the wind farm pressure field in the realistic GW case with the idealized rigid lid case. The rigid lid approximation retains some of the features of the atmospheric problem but allows us to derive simple theorems and closed form solutions that clarify the cause, properties and impact of the pressure field. The pressure disturbance decays inversely as the first power of the distance upstream.


2.   The Gravity Wave (GW) model and the Rigid Lid (RL) limit

Our method for computing the response to wind farm drag forces uses a 2-layer stratified hydrostatic Gravity Wave (GW) model solved with Fast Fourier Transforms (FFT). This model consists of a "turbine layer" from which momentum is removed
by specified drag forces and an overlying density-stratified layer that responds to vertical displacement and whose hydrostatic pressure field acts back on the turbine layer (Smith 2010, 2022). The governing equations for the model include a Rayleigh restoring force to decay the farm wake. The special rigid lid solution is found by increasing the density stratification until the interface atop the lower layer does not deflect.

We first ran the two-layer GW model with the "realistic" parameters shown in Table 1. The model's two stability parameters
are the reduced gravity g' of the inversion and Brunt-Vaisala frequency N of the troposphere given by

$$g' = g\frac{\Delta\theta}{\theta} = 0.1 \, ms^{-2} \quad \text{and} \quad N = \sqrt{\frac{g}{\theta}\frac{d\theta}{dz}} = 0.01 \, s^{-1} \tag{1}$$

where  θ is the potential temperature.  We set the Rayleigh restoring coefficient C to a fairly small value so the wake recovery is slow, but fast enough to prevent wrapping. To compute the average turbine drag force, we suppose a ratio of rotor disk area to farm area of DAR=0.0077 and a turbine thrust coefficient of $C_T = 0.75$. With a wind speed of $U = 10ms^{-1}$ and turbine
layer depth of H=400m, the wind farm drag per unit air mass is

$$F = \frac{DAR*C_T*U^2}{2H} = \frac{(0.0077)(0.75)(10^2)}{2(400)} = 0.0007218 \, ms^{-2} \tag{2}$$

For illustration, we chose horizontal farm dimensions a=b=7000m. The total drag on the farm is then

$$Drag = \rho \cdot a \cdot b \cdot H \cdot F \approx 17 \cdot 10^6 \, Newtons \tag{3}$$

A few output parameters from this reference run are given in Table 2.  The inversion is displaced upward by 11.8 meters over
the farm and there is a 2.38Pascal pressure difference across the farm.

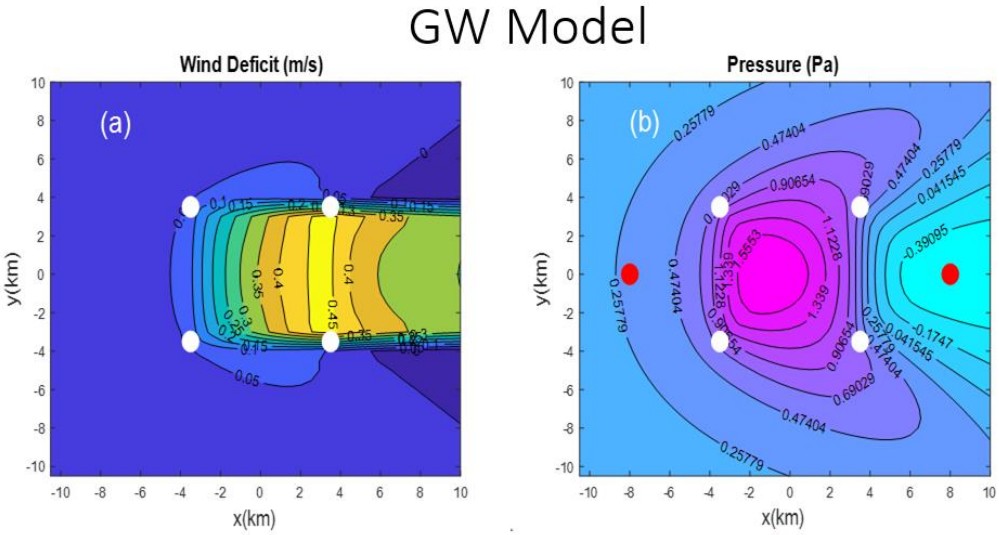

Figure 1: Zoom of the disturbance from a 7km by 7km wind farm from the realistic gravity wave (GW) model: (a) wind speed deficit and (b) pressure. Airflow is from left to right. White dots mark the corners of the farm. In (b), the red dots are pressure sampling points. The full domain is 200km by 200km.

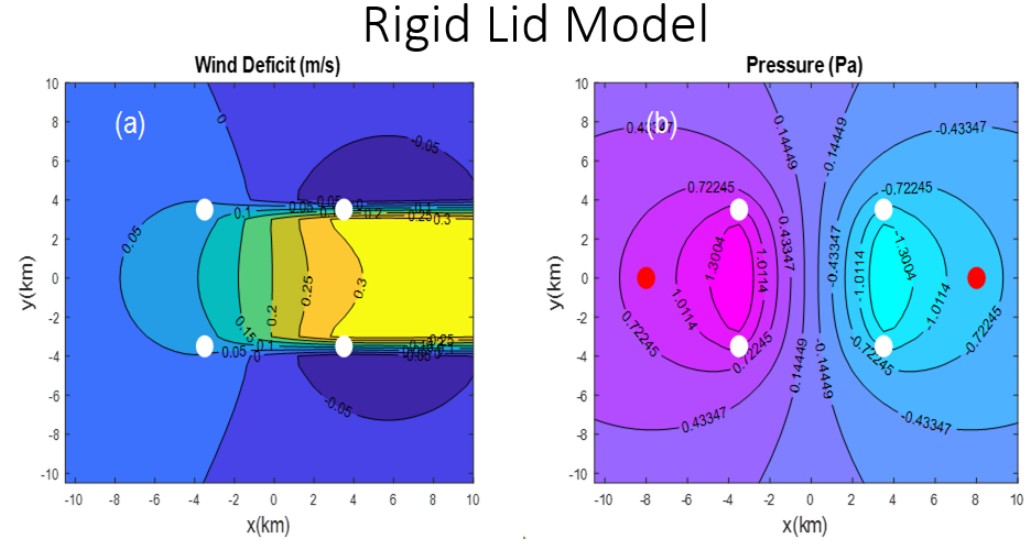

Figure 2: Zoom of the disturbance from a 7km by 7km wind farm from the idealized rigid lid (RL) case: (a) wind speed deficit and (b) pressure. Airflow is from left to right. White dots mark the corners of the farm. In (b), the red dots are pressure sampling points. The full domain is 200km by 200km.





Table 1: Parameters of the reference GW model

| Parameter | Symbol | Units | Value |
|---|---|---|---|
| Ambient Wind speed | U | $ms^{-1}$ | 10 |
| Layer depth | H | $m$ | 400 |
| Applied drag force | F | $ms^{-2}$ | 0.0007218 |
| Farm Drag | Drag | $Nt$ | 17E06 |
| Interface reduced gravity | g' | $ms^{-2}$ | 0.1 |
| Tropospheric Stability | N | $s^{-1}$ | 0.01 |
| Rayleigh restoring coefficient | C | $s^{-1}$ | 0.00033 |
| Farm size | a, b | $km$ | 7 by 7 |
| Grid size | dx, dy | $km$ | 0.5 by 0.5 |

To investigate the influence of atmospheric stability (1), we ran the GW-FFT model several more times, first with the two stability parameters g'=N=0. When there is no stability, the turbine drag slows the airstream and displaces the top of the turbine layer upwards, but no pressure disturbance is generated.  Any pressure disturbances arising from vertical accelerations are neglected here with the hydrostatic assumption.

We then increased each stability parameter (1) from zero. The vertical displacement of the interface decreased towards zero
and the pressure field increased from zero. Other model output values changed only slightly.  The maximum wind speed deficit decreased slightly from $0.445ms^{-1}$ to $0.323ms^{-1}$ in the rigid lid limit.  The average relative speed deficit over the farm ($\gamma = \bar{u}/U$) decreased slightly from $\gamma = 0.0226$ to $0.0195$.

One striking aspect of Table 2 is that the g' series and the N series of runs approach the same "rigid lid" limit.  The trends are smooth for the N series but the g' series of runs shows a singularity when the Froude Number $Fr = U/\sqrt{g'H} \approx 1$.
Ultimately, increasing either type of stability takes us to the same rigid lid solution with finite wind deficit, pressure difference and a vanishing vertical displacement.  When $N = 0$, the displacement approaches zero as $1/g'$ and when $g' = 0$ it approaches zero as $1/N$.

The planform patterns of the gravity wave (GW) and rigid lid (RL) solutions are compared in Figures 1 and 2.  The wind speed deficit patterns (Figs 1a, 2a) show the wake caused by the farm drag but also show the influence of the pressure fields. Both
show upstream deceleration, stronger in the RL case, and lateral regions of accelerated flow downwind of the farm. The wind speed deficit patterns over the farm are different too due to pressure forces acting on the flow. The pressure fields (Figs 1b,2b)



show an upwind maxima and downwind minima of approximately similar magnitude. The RL case however has these extrema shifted upwind and the whole field is exactly anti-symmetric.


Table 2: Increasing stability towards the rigid lid limit

| g' | N | Maximum Displacement | Maximum Deficit | Gamma $\gamma = \bar{u}/U$ | $\Delta p$ (9) | A (15) |
|---|---|---|---|---|---|---|
| $ms^{-2}$ | $s^{-1}$ | $m$ | $ms^{-1}$ | - | $Pa$ | $Pa \cdot m$ |
| 0.1* | 0.01* | 11.7 | 0.468 | 0.0315 | 2.38 | 2335 |
| 0 | 0 | 18 | 0.445 | 0.0226 | 0 | 0 |
| 0.05 | 0 | 21 | 0.539 | 0.0272 | 1.33 | 0 |
| 0.1 | 0 | 18 | 0.589 | 0.0236 | 2.57 | 0 |
| 0.2 ** | 0 | 21.6 | 0.682 | 0.0507 | 7.06 | 0 |
| 1 | 0 | 1.72 | 0.307 | 0.0196 | 3.94 | 8302 |
| 10 | 0 | 0.135 | 0.32 | 0.0194 | 3.24 | 6821 |
| 100 | 0 | 0.0132 | 0.323 | 0.0194 | 3.18 | 6702 |
| 1000 | 0 | 0.0013 | 0.323 | 0.0194 | 3.18 | 6691 |
| 1E06 | 0 | 1.3E-06 | 0.323 | 0.0194 | 3.18 | 6689 |
| 0 | 0 | 18 | 0.445 | 0.0226 | 0 | 0 |
| 0 | 0.005 | 13.9 | 0.444 | 0.0247 | 0.595 | 906 |
| 0 | 0.01 | 11.9 | 0.432 | 0.0257 | 1.09 | 1754 |
| 0 | 0.02 | 8.8 | 0.403 | 0.0259 | 1.81 | 3132 |
| 0 | 0.1 | 2.4 | 0.335 | 0.0222 | 2.99 | 6019 |
| 0 | 1 | 0.25 | 0.324 | 0.0197 | 3.17 | 6646 |
| 0 | 10 | 0.025 | 0.323 | 0.0195 | 3.18 | 6686 |
| 0 | 100 | 0.0025 | 0.323 | 0.0195 | 3.18 | 6689 |
| 0 | 1E06 | 2.5E-07 | 0.323 | 0.0194 | 3.18 | 6689 |



*Reference GW case, $** Fr \approx 1$


3. The Harmonic Pressure Field

An understanding of the pressure field in the GW case and the RL limit requires an analysis of the linearized steady momentum

equation for the turbine layer

$$U\left(\frac{\partial u}{dx}\right) + V\left(\frac{\partial u}{\partial y}\right) = -\rho^{-1}\left(\frac{\partial p}{\partial x}\right) + F_x - Cu \qquad (4a)$$

$$U\left(\frac{\partial v}{\partial x}\right) + V\left(\frac{\partial v}{\partial y}\right) = -\rho^{-1}\left(\frac{\partial p}{\partial y}\right) + F_y - Cv \qquad (4b)$$

where $\vec{F}$ is the turbine drag (2), $\vec{U}$ is the ambient wind, $\vec{u}$ is the drag-induced perturbation wind and C is the Rayleigh restoring

coefficient (Smith 2022). The pressure field p(x,y) is derived using the hydrostatic assumption. When this equation is solved

for the perturbation wind, the scalar wind deficit is computed from $Deficit = -(\vec{U} \cdot \vec{u})/|\vec{U}|$ and its area integral is the

Total Deficit

$$TD = \iint Deficit(x, y)\, dxdy \qquad (5)$$

Taking the dot product of (4) with the ambient wind $\vec{U} = (U, V)$ and integrating over the whole domain relates TD to the

turbine drag (Smith 2022)

$$TD = \frac{-1}{|\vec{U}|C} \iint \vec{U} \cdot \vec{F}(x, y)\, dxdy \qquad (6)$$

Because the pressure field p(x,y) decays at infinity, it does not influence TD but just alters the spatial distribution of

$Deficit(x, y)$.

In the rigid lid solution, the pressure field $p(x, y)$ is a harmonic function almost everywhere. A Harmonic function is one

which satisfies Laplace's Equation $\nabla^2 p = 0$. To prove this hypothesis, we apply the divergence operator to (4) giving

$$U\left(u_x + v_y\right)_x + V\left(u_x + v_y\right)_y = -\rho^{-1}\left(p_{xx} + p_{yy}\right) + F_{x,x} + F_{y,y} - C\left(u_x + v_y\right)$$

or $$\vec{U} \cdot \nabla(\nabla \cdot \vec{u}) = -\rho^{-1} \nabla^2 p + \nabla \cdot \vec{F} - C(\nabla \cdot \vec{u}) \qquad (7)$$

In 2-D non-divergent flow, the LHS of (7) is zero and

$$\nabla^2 p = \rho \nabla \cdot \vec{F} \qquad (8)$$

so the pressure field is a Harmonic function except at the windward and leeward edges of the wind farm where $\nabla \cdot \vec{F} \neq 0$..



To illustrate the Harmonic property of p(x,y), we show the Laplacian of the pressure field for the reference GW case in Fig. 3a and the Rigid Lid case in Fig. 3b. They differ in important details. In Fig. 3a, $\nabla^2 p = 0$ is violated over most of the field in a complicated pattern while in Fig. 3b it is violated only over the farm front and back edges, in agreement with (8). The Laplacian in Fig 3 was computed in Fourier space with $\hat{\Delta}(k, l) = -(k^2 \hat{p}(k, l) + l^2 \hat{p}(k, l))$ and then inverted.

Recall that a harmonic function has no local maxima or minima and therefore only takes on values that are between the
boundary values. As p(x,y) decays at infinity, the pressure would therefore vanish were it not for these two local non-Harmonic extrema. Thus, these two extrema in Fig 3b, "support" or "cause" the pressure field seen in Fig 2b.

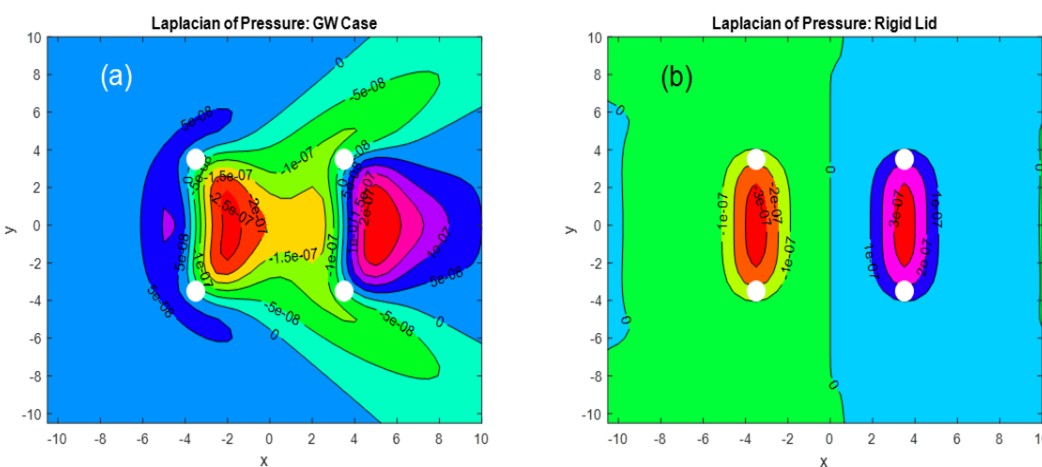

Figure 3: Laplacian of the pressure with units $Pa \cdot m^{-2}$. a) Reference GW case, b) Rigid Lid case. Airflow is from left to right. White dots mark the corners of the farm. A low pass filter has been applied to (b).


5. Role of the pressure field

The two pressure fields, GW and Rigid Lid, are compared along the centerline in Figs. 4a,b. Both transects have an upwind maximum and downwind minimum. The GW pressure field (Fig 4a) is smoother with a maximum over the farm and a smaller minimum in the wake. In the rigid lid case (Fig 4b), the pressure maximum and minimum points are equal in magnitude and
shifted upstream slightly to the farm edges. In both cases, the air decelerates as it approaches the farm under the adverse pressure gradient. The linearized Bernoulli equation $Uu(x) = -\rho^{-1} p(x)$ is approximately valid upwind, so as the



pressure rises the wind speed drops. There is also an adverse pressure gradient downwind of the farm. Overall, the pressure field smooths out the velocity field by spreading the deceleration up and down wind.

A key feature of the rigid lid solution is the opposing friction (F) and pressure gradient force (PGF) over the farm. As the

pressure is a nearly linear function there, we define

$$\Delta p = Max(p) - Min(p) \tag{9}$$

so

$$PGF = -\rho^{-1}\left(\frac{dp}{dx}\right) \approx \Delta p/\rho a \tag{10}$$

Using values from Tables 1 and 2, the non-dimensional force ratio is

$$\frac{PGF}{F} = \frac{(3.18Pa)}{(1.2kg \cdot m^{-3})(7000m)(-0.0007218ms^{-2})} = -0.52 \tag{11}$$

Thus, in this case, the favorable pressure gradient cancels 52% of the turbine drag over the farm. The magnitude of this ratio increases with aspect ratio $AR = b/a$.

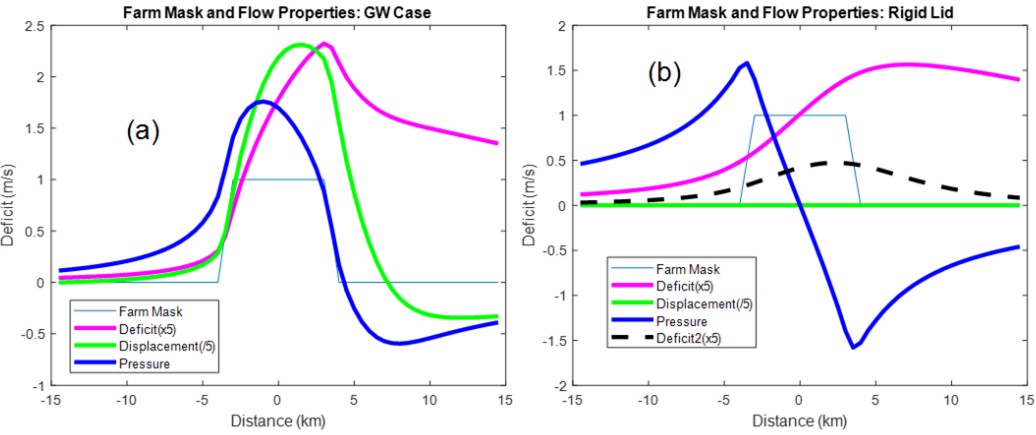

Figure 4: Centerline properties of the farm disturbance including the farm mask, wind speed deficit ( x5 m/s), interface displacement (/5 m) and pressure (Pa): a) Reference GW case, b) Rigid Lid case. In (b), the dashed line is the wind speed deficit with a larger Rayleigh restoring coefficient ($C = 0.0033 \ s^{-1}$). The pressure is unchanged. Airflow is from left to right.


6. The cause of the pressure field





Further insight into the cause of the pressure field in the rigid lid case comes from noting that pressure is insensitive to the Rayleigh restoring force coefficient C in (4). This fact is seen in (8) but we illustrate it in Fig 3b. The dashed curve in Fig 3b,

shows the wind speed deficit where we increase the coefficient tenfold to $C = 0.0033 \, s^{-1}$, so that the wake decay length (L=U/C) is 3km instead 30km. In Fig. 3b, the wind speed deficit is dramatically reduced while the pressure field is unchanged. This independence of the pressure field from C is a unique feature of the Rigid Lid case and not found in the more general GW case where the Rayleigh force is divergent.

In incompressible or non-divergent flow, the role of pressure is to maintain the non-divergent property of the flow. As the

turbine force field $\vec{F}(x,y)$ is divergent, the pressure field must arise instantly to prevent any flow divergence. That is the meaning of (8). The Rayleigh force in this case is non-divergent, so it does not create any pressure field.

7. The far-field pressure

Equation (8) is the Poisson Equation where the scalar $\rho \nabla \cdot \vec{F}$ is the equivalent of a "point charge" in an electrostatic analogy.

If we define $B(x,y) = \rho \nabla \cdot \vec{F}$ then the general solution to (8) using a Green's function is

$$p(x,y) = \left(\frac{1}{4\pi}\right) \iint \ln((x-x')^2 + (y-y')^2) \cdot B(x',y') \, dx'dy' \qquad (12)$$

While the logarithm function in (12) diverges at infinity, (12) itself is well behaved because $\iint B dx dy = 0$. If we lump the front and back edge contributions into two delta functions,

$$B(x,y) \approx \rho F b (\delta\left(x + \frac{a}{2}, y\right) - \delta\left(x - \frac{a}{2}, y\right)) \qquad (13)$$

then from (12) for r>>a, we obtain asymptotically the dipole

$$p(x,y) \approx \frac{-\rho F a b x}{2\pi r^2} = -A\left[\frac{x}{r^2}\right] \qquad (14a)$$

where $r = \sqrt{x^2 + y^2}$ and the constant

$$A = \left(\frac{1}{2\pi}\right) \rho \cdot F \cdot a \cdot b \qquad (14b)$$

This dipole formula (14) is consistent with the pressure field pictured in Fig 2b. The isobars for (14a) are circles touching each other at the origin. Thus, (14a) satisfies $p(x,0) = p(x/2, \pm x/2)$). On the 45 degree lines ($y = \pm x$), $\partial p / \partial x = 0$; so the isobars are parallel to the x-axis there.

In the present computation (Table 1), the drag force is $F = 0.0007218 \, ms^{-2}$ so we predict from (14b) that $A = 6754 Pa \cdot m$. We checked this prediction against our computed pressure field (Fig 3) by using the pressure at distance $d = $

$8km$ upstream from the farm center. Using (14a)



$$A = p(x = -d, y = 0)(d) = 6689 Pa \cdot m \tag{15}$$

(see Table 2). The small 1% difference between these two A values verifies our solution. The 1% difference arises from the fact that 8km is not far enough upstream to be in the "far field".

The Green's Function method with two delta-functions (12,13) can also be used to find the pressure field near the farm center. The result is

$$p(x, y) \approx -2 \frac{\rho F b x}{a \pi} \tag{16}$$

The non-dimensional force ratio for a=b is then

$$\frac{PGF}{F} \approx -\frac{2}{\pi} \approx -0.64 \tag{17}$$

roughly similar to the computed value in (11).

## 8. Alternate derivation of the drag induced pressure dipole

In the previous section, we used a Green's function to derive the far-field pressure dipole (14). We now re-derive this formula using a physical volume-conservation argument. When the farm drag slows the flow, it creates a volume flow deficit (Q) in the wake. A farm with downwind dimension "a" with drag Force (F) per unit mass (units $ms^{-2}$) will create (from 4) a wake with speed deficit $Deficit = F \cdot a/U$. The lost volume flux in the wake is

$$Q = Deficit \cdot b \cdot H = F \cdot a \cdot b \cdot H/U \tag{18a}$$

or using (3)
$$Q = \frac{Drag}{\rho U} \tag{18b}$$

with units $m^3 s^{-1}$. We balance the volume budget by adding an equal point source at the origin. Confined to a layer of depth H, the velocity field from a point volume source Q is

$$u = (\frac{Q}{2\pi H})(\frac{x}{r^2}) \text{ and } v = (\frac{Q}{2\pi H})(\frac{y}{r^2}) . \tag{19}$$

The radial speed is $u_r = \sqrt{u^2 + v^2}$ so the volume flow is $Q = 2\pi r H u_r$ .

If the mean flow U is added to the source flow (19), the total fluid speed at each point is

$$S^2 = (U + u)^2 + (v)^2 . \tag{20}$$

This combined flow is equivalent to the familiar Rankine Half-body of width $W = \frac{Fab}{U^2}$. A similar approach was used by Gribben and Hawkes (2019) for a single turbine. In the absence of dissipation, Bernoulli's equation gives the pressure anomaly at each point

$$p(x, y) = -(\frac{1}{2}) \rho (S^2 - U^2) \tag{21}$$

Linearizing with (20,21) gives a dipole pressure pattern in the far-field





$$p(x,y) \approx -\rho U\left(\frac{Q}{2\pi H}\right)\left(\frac{x}{r^2}\right) = -A(x/r^2) \qquad (22)$$

where

$$A = \left(\frac{1}{2\pi}\right)\rho \cdot F \cdot a \cdot b \qquad (23)$$

in agreement with (14). If the total farm drag has been computed in Newtons, then using (3) the pressure coefficient

$$A = \left(\frac{1}{2\pi H}\right) Drag \qquad (24)$$

where H is the depth of the layer into which the Drag has been applied. The pressure coefficient A has units $Pa \cdot m$. If the farm is not rectangular, the product $a \cdot b$ in (23) can be replaced with the farm area.

5. Blocking and Deflection

As the RL source function expression (19) provides good estimates of the far field pressure, we can use it to estimate airflow blocking and deflection. For upstream blocking, the wind disturbance will decay inversely with distance upwind. At the front edge of the farm, we evaluate (19) to give

$$u\left(x = -\frac{a}{2}, y = 0\right) = \left(\frac{F \cdot a \cdot b}{2\pi U}\right)\left(\frac{x}{r^2}\right) = -\frac{F \cdot b}{\pi U} \qquad (25)$$

The small pressure reduction and wind speed maxima near the downwind farm corners (Fig 1) can also be explained with these formulae (22,25).

The upwind pressure field deflects the airflow to the left and right. The maximum lateral speed is located near the farm edge at $x = 0, y = b/2$. From (19),

$$v\left(x = 0, y = \frac{b}{2}\right) = \left(\frac{F \cdot a \cdot b}{2\pi U}\right)\left(\frac{y}{r^2}\right) = \frac{F \cdot a}{\pi U} \qquad (26)$$

In the present example with a=b (Table 1), the magnitudes of u and v are both $0.16 ms^{-1}$. Potential errors in (25, 26) come from using the far field formulae too close to the farm and the influence of Rayleigh friction.

6. Determining total farm drag from pressure measurements

The direct link between farm drag and far-field pressure dipole (24) allows us to determine total farm drag with a pair of pressure measurements. If pressure sensors are located a distance "d" upwind and downwind of the farm center, then the difference in pressure between those two sensors $\Delta P_M$ gives the pressure dipole coefficient using (14 or 22)

$$A = \Delta P_M \cdot d/2 \qquad (27)$$

From A, the total farm drag is found using (24)

$$Drag = 2\pi HA \qquad (28)$$





In the rigid lid case (Fig 3b) the pressure values 8km upstream and downstream are $p = \pm 0.84 Pa$ so $\Delta P_M = 1.68 Pa$ . Using (27,28), we obtain $A = 6720\ Pa \cdot m$ and the total farm drag is $Drag \approx 17 \times 10^6 Nt$ in agreement with the specified drag in Table 1.

In the reference GW case (Fig 3a), the upstream and downstream pressure values are $p = 0.292 Pa$ and $p = -0.607 Pa$ so $\Delta P_M = 0.899 Pa$ . Using these GW case values in the rigid lid formulae (27,28) gives $Drag \approx 9 \times 10^6 Nt$. Thus the error in (27, 28) is large, but a measured $\Delta P_M$ still provides a useful lower bound on the farm Drag. If more accuracy is needed, use the linear GW model or a full-physics mesoscale model.


## 11. Discussion

The nature of airflow deceleration by a wind farm depends on the atmospheric static stability. With no stratification or other vertical "confinement", the slowed turbine layer will thicken and push tropospheric air upwards. Due to the lack of stratification, this upward displacement will not generate a hydrostatic pressure disturbance. With no pressure gradient forces,

the airflow will not be decelerated until it reaches the farm and there will be no lateral deflection. Over the farm, the turbine drag will act directly to slow the flow. When moderate stability is present, the upward displacement will create pressure anomalies that act on the lower layer. These pressure gradients can slow upstream air and deflect the air laterally. The computation of the pressure field requires the treatment of gravity waves. When the stratification is very strong, we approach the rigid lid limit. Little or no vertical displacement occurs. We can compute the pressure field directly from the non-divergent

assumption, without having to consider gravity waves. The pressure field is created naturally to prevent flow divergence. It blocks and deflects the flow and partly cancels the turbine drag force over the farm.

The rigid lid approximation maintains some properties of the full gravity wave (GW) solutions while offering closed form expressions that deepen our understanding of the wind farm pressure disturbance. In the real atmosphere, the inversion strength is only about $g' = 0.1\ ms^{-2}$ and the tropospheric stability is about $N = 0.01 s^{-1}$. With these values, the interface

may be significantly displaced but the confinement is sufficient that already some of the rigid lid characteristics appear (Figs 1,2,4). The stability values need to be an order of magnitude larger before the rigid lid approximation becomes quantitatively accurate.

The rigid lid solution provides an alternative way to understand the upwind pressure field and upstream blocking. In the GW case, we favor an explanation that uses the upwind propagation of shallow layer waves (when Fr<1) and the upwind tilt of

deep internal waves to explain the upwind bulge of high pressure. Ascending air upstream in these waves give positive density anomalies that hydrostatically generate high pressure below. The rigid lid case however, favors an explanation related to simple vertical confinement. The condition $\nabla \cdot \vec{u} = 0$ requires a pressure field to counteract the divergent field of turbine drag (8). The wind farm pressure dipole was derived in two ways: first, solving Poisson equation with delta functions and second, applying a volume source proportional to total farm drag.



In both cases, GW and RL, pressure forces act to smooth out the deceleration of the wind by the farm. They reduce the deceleration over the farm with a favorable pressure gradient and add deceleration zones upwind and downwind with an adverse pressure gradient. They can also produce small areas of accelerated airflow to the left and right of the farm.

To understand wind farm dynamics, more effort should be made to measure the pressure field near wind farms and to diagnose pressure fields in numerical models. As the pressure field arises directly from the divergence of the turbine drag force and is s

proportional to the farm drag, pressure measurements may provide a check on our drag estimates. If pressure differences are larger than expected, it may indicate that we are underestimating the total drag on wind farms; including the change in ocean surface wind stress caused by turbine induced boundary layer turbulence. According to (6), one could measure the Total Deficit (TD) to determine Drag but that method requires knowledge of the wind restoring force coefficient ( C ). Instead, (27, 28) might provide a more direct Drag estimate.


Acknowledgements

I appreciate useful conversations with Brian Gribben, Graham Hawkes, Neil Adams, Xiaoli Guo Larsen, Jake Badger, Jana Fischereit and Idar Barstad.

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
