# Peer review of "The wind farm pressure field"

_Wind Energy Science, 2023_

## Referee Comment (RC1)

The paper analyses the perturbation of the atmospheric pressure field due to wind farms based on a linear, hydrostatic, two-layer model. Two scenarios are considered. The first scenario corresponds to a realistic case in which the atmosphere above the wind turbine layer is stably stratified. In this scenario, flow divergence in the wind turbine layer leads to a vertical displacement of the layer above and hence supports the formation of atmospheric gravity waves. The second scenario is the limit in which the stable stratification is so strong that the vertical displacement of the atmosphere above the wind turbine layer is zero so that the stratification effectively acts as a rigid lid. This second, asymptotic case is analyzed in depth to better understand the role, cause, and impact of the pressure field. For this case, the paper offers insight into how and why the atmospheric pressure field is perturbed by wind farms, and simple expressions for the pressure and velocity in the far field are derived. I believe the paper addresses an important aspect of wind farms that is not well understood, and the insights it presents could therefore potentially be very useful to the wind energy community. However, I miss a deeper discussion on how the findings of the rigid lid case translate back to the first realistic scenario. Moreover, while the theoretical analysis is sound, some intermediate steps and results need to be made more clear to better guide the reader. Please find a detailed list of comments and suggestions.

**Main comments**

1. At the end of the introduction I miss an outline of the paper. As no outline is presented, it is currently not clear how the paper is structured. This makes it harder to understand the relevance and significance of the paper.

2. The comparison of the case with atmospheric gravity waves versus a rigid lid scenario is also interesting in light of the two main approaches for large-eddy simulations of wind farms that can be found in the literature, i.e., either resolving the atmospheric boundary layer and part of the free atmosphere (which supports the formation of gravity waves) or using a pressure-driven boundary layer with a rigid lid condition. A short literature survey and a quantitative discussion on how different the two approaches are would therefore be useful.

3. The description of the model in section 2 is too limited. Part of the governing equations are in fact shown in section 3, so why not present them when introducing the model? Furthermore, on line 109 it is stated that "the pressure field p(x,y) is derived using the hydrostatic assumption," but it is not clear to me what is meant by this.

4. Sections 3-10 all seem to focus on the rigid lid case, but the link back to the realistic case is missing a bit. What do the conclusions for the rigid lid case mean for the more realistic case? How different are the results when the inversion is not a perfect rigid lid?

5. I have the impression that some of the theoretical results are in line with previous findings in the literature. For example, equation 25 shows that upstream flow blockage increases with farm width, and this has been found before (e.g. Allaerts & Meyers (2019). Same for the decay of the perturbation away from the farm (see eq 19 and 22), has this behavior been observed before (not sure myself)? It would be worthwhile to tie obtained results with what has been found before.

**Specific comments**

1. Line 19: "... the wind slowing by these farms ... . This issue has an extensive literature, ..." I find this statement about the wind slowing a bit vague. Do you refer to the upstream flow deceleration, the wind deficit behind individual turbines, slowing down of the atmospheric boundary layer above the wind farm? All of these? Please make this more clear.

2. Line 25: "In a stably stratified atmosphere, ..." I think this statement can be misleading for the reader. Gravity waves only form in regions where the flow is stably stratified, but you can get gravity waves also above neutral and unstable boundary layers. The statement could be misinterpreted as if gravity waves only occur when the atmospheric boundary layer is stably stratified (i.e. when the surface heat flux is negative).

3. Line 28: I believe the term used by Bleeg (and many others) is blockage rather than blocking.

4. Line 29: "... over the farm, it can fight back against the turbine drag, ... . Finally, it alters the recovery of the wake." Have these effects been observed in the literature? Please cite relevant studies.

5. Line 33: "According to Gribben and Hawkes (2019), the local non-hydrostatic pressure disturbances decay inversely ..." Please clarify whether this result is for a single turbine or for a farm.

6. Line 34: "The farm-generated hydrostatic pressure disturbance is more far-reaching." What do you base this statement on? Evidence from literature (if so, add references), or is this based on your own findings (then say something like "as will be shown in this study, ...").

7. Line 43: Is there a specific reason why the model is limited to hydrostatic gravity waves?

8. Line 53: Specify what you mean by "wrapping"

9. Line 54: Please define "DAR" more clearly (I assume rotor disk area to covered surface area, where the latter is assumed to be $s_x s_y D^2$).

10. Line 86: "When N=0, the displacement approaches zero as $1/g'$ and when $g'=0$ it approaches zero as $1/N$." This relationship would be more clear when plotted in a figure.

11. Improve caption of table 2. Currently, it is not clear what is listed and how some of the parameters are defined (e.g. for the definition of Gamma one needs to refer to the main text).

12. Line 109: "When this equation is solved for the perturbation wind, the scalar wind deficit is computed from ..." This transition is too fast, I needed to look up the definition in Smith (2022) to understand that the definition results from a linearization. Please explain more clearly how the scalar wind deficit is obtained.

13. Line 116: "Because the pressure field p(x,y) decays at infinity, it does not influence TD ..." Again, going a bit fast. The pressure term vanishes because of the divergence theorem and the fact that p decays at infinity, but for this you also need the continuity equation to go from $\mathbf{U} \cdot \nabla p$ to $\nabla \cdot \mathbf{U} p$.

14. Line 122: "In 2-D non-divergent flow ..." The assumption of non-divergent flow can be made because of the rigid lid case, or does this also hold in the GW case? Please clarify.

15. Figure 3: Why do you apply a low pass filter to (b)? What high-frequency content are you filtering out? Noise due to the low resolution?

16. Line 146: "The linearized Bernouilli equation ... is approximately valid upwind, ..." Upon what is this statement based? Did you check whether this approximation is valid?

17. Line 156: "The magnitude of this ratio increases with aspect ratio ..." Not clear to me why this would be the case. Did you try other aspect ratios?

18. Line 170: "... the wake decay length (L=U/C) ..." Is this wake decay length defined in literature? If not, it should be made clear why the wake decay length is defined like this.

19. Line 174: "In incompressible or non-divergent flow, ..." In the GW case, the flow is also incompressible, but flow divergence is allowed and leads to inversion displacement. Does that mean that the role of the pressure as you describe only holds for incompressible *and* non-divergent flow?

20. Line 281: "The stability values need to be an order of magnitude larger before the rigid lid approximation becomes quantitatively accurate." What criteria is this statement based on? When is the rigid lid approximation considered quantitatively accurate? This needs more explanation.

21. Line 296: "... including the change in ocean surface wind stress caused by turbine induced boundary layer turbulence." Not very clear, and also not sure why this is relevant (not discussed anywhere else in the paper).

**Technical comments**

1. Line 28: so call "Blocking" → so call*ed* "Block*age*" (see also earlier comment)

2. Table 2, entries for parameter A for cases g'=0.05, 0.1, and 0.2 with N=0: The value for A is exactly equal to zero. Is this correct or is this a typo?

3. Section 4 is missing?

---

## Referee Comment (RC2)

The manuscript focuses on wind-farm-induced pressure fields, highlighting their important influence on flow through a wind farm. Historically, pressure has not been a significant consideration among modelers and analysts of wind farm flows, but this "friction-only" approach, as I believe the author put it in a previous paper, has fallen from favor over the last few years. The wind energy community is paying much more attention to inviscid influences now, and research that can provide more insight into these influences are very welcome. I believe that this paper falls into this category of research, but before publication I would recommend a number of changes. The following review starts with the **main comments**, listed roughly in order of priority, followed by a list of **minor comments**.

**Main comments**

**1) Reliability and significance of the rigid lid results**

The agreement between the GW and RL results is, in my view, poor. For example, internal wake recovery due to the favorable pressure gradient would be much different in the GW case than the RL case as illustrated in Figure 4 (the favorable gradient is much stronger in the RL case). The author acknowledges the differences between the GW and RL results on line 281: "The stability values need to be an order of magnitude larger before the rigid lid approximation becomes qualitatively accurate." Although it has its own simplifications, the GW model clearly offers a better representation of the interaction of a wind farm with the atmosphere than the RL model. Given this and the significant differences in the results, I am struggling to understand how I should glean credible insight into wind farm flows from the RL results. Yes, there are some qualitative similarities between the GW and RL results, perhaps most notably the favorable pressure gradient through the wind farm. However, I think even a very basic first-principles-based analysis would arrive at the conclusion that there must be a wind-farm-scale favorable pressure gradient through the wind farm.

As stated in the introduction, the paper aims to use the RL results to "clarify the cause, properties and impact of the pressure field." Could we not simply get this clarification from the GW model? Perhaps I missed something in the author's point about the causes of the pressure field as summarized in the section 6. When I first read the part about the pressure field responding to divergence in the turbine force field, I assumed it was in the context of the RL case only, after all equation 8, which is only valid for the RL case, is highlighted here. Is the author also making the point that the divergence of the force field influences the pressure field in the GW model? If so, it would be better to be more explicit about this point.

All that said, I did find the comparison between the RL and GW to be compelling. A few years ago, the industry started to use simple potential flow models, without consideration of stratification, to represent the impact of inviscid effects on wind farm flows. With the growing recognition that these models fall short and that stable stratification and gravity waves are important influences (due in no small part to the author's pioneering work in this area), many potential flow modelers are now trying to use a rigid lid to represent the impact of stratification on the wind farm. Your results suggest to me that a rigid lid does not offer a good approximation of the impact of stratification and gravity waves on wind farm flows and is therefore likely a significant source of error in models that use it instead of a more complete representation of the stably stratified atmosphere. If the author agrees with this view, *I feel it should be highlighted in the paper*. The matter has substantial practical implications for the industry, as potential flow codes, either without any modeling to represent

stable stratification or a rigid lid to represent its impact, are being used today to support the development of large wind farms, many of which are valued at well more than a billion dollars.

**2) Wind-farm-induced pressure field without stratification**

There are at least two parts of the manuscript that suggest that there would not be a wind-farm-induced pressure field without flow stratification. For example, the very first sentence of the abstract reads, "The disturbed atmospheric pressure near a wind farm arises from the turbine drag forces in combination with vertical confinement associated with atmospheric stability." Another example can be found in the lines 267-270, which can be (mis)read to mean that with no stratification there will be no pressure gradient forces. I think it is a fair assumption that the author is aware that there can be wind-farm-induced pressure disturbances without stratification. For example, line 33 of the manuscript ("the local non-hydrostatic pressure disturbances decays…") is consistent with this assumption.

The disturbed hydrostatic pressure field clearly has a significant impact on the wind farm flow. It clearly has a much larger impact on the flow farm upstream of the wind farm and the wind-farm-scale favorable pressure gradient through the wind farm as compared "local non-hydrostatic pressure disturbances". I'm just wondering if the highlighted sentences could be changed slightly to make them more accurate. I would be OK with just adding "primarily" in front of "arises" in the first sentence of the abstract (although local non-hydrostatic pressure disturbances, of course, have a big impact very close to the wind turbines). I don't have a specific edit for lines 267-270, but I believe they should be edited to avoid misunderstandings, as deceleration upstream of the wind farm would still happen without stratification or other confinement, just to a much smaller extent.

**3) More description**

In my opinion, readers would benefit from more detailed descriptions in the paper. At a number of points, I felt left to try to figure out the definition of a term or how an aspect of the model worked. Here are some examples:

- Please explicitly define DAR. I wasn't sure until I plugged in some numbers myself.
- The reader is also left to figure out the specific definition of F(x,y). It appears to be uniform in the wind farm area, pointing in the opposite direction of the flow, and zero outside. One can figure this out, but it would be better to be explicit here.
- What is "wrapping"?
- I think the reader would benefit from more description of the model. You reference Smith 2010 and 2022, and they are indeed very helpful when it comes to understanding the model. However, I think the reader would appreciate a more information about the model within the manuscript.

**4) Utility of measuring pressure in the field**

The last section recommends measuring pressure around a wind farm in part as a means of estimating the magnitude of the wind farm drag. This would, of course, be a very rough approximation. The turbine SCADA data in conjunction with the power and thrust coefficient curves for the model would provide a much more reliable estimate of the total wind farm drag than pressure measurements in combinations with equations 27 and 28.

**Minor comments**

Line 28: 'so call "Blocking"'. "Blockage" is the more commonly used term in the wind energy community. It is also the term used in the paper. "so-called" should probably modify "blockage" rather than "so call".

Line 80: "the pressure field increased from zero." I think this phrasing can be improved upon. I'm pretty sure I understand what the author means when he writes that the "pressure field increased", but maybe the "pressure disturbances increased" would be better.

Line 122: It may be worth pointing out that the right-most term in the equation is also zero.

Lines 172 and 176: The sentence starting "The Rayleigh force in this case is non-divergent…" on line 176 appears to repeat a point just stated on line 172 in the sentence starting "This independence of the pressure field…" Or perhaps I've missed a subtlety. If not, perhaps the second sentence should be deleted.

Line 294: There is a typo at the end of the line between "is" and "proportional"

---

## Author Comment (AC2)

Responses to Reviewers Comments

WES-2023-56 | Research article
Submitted on 22 May 2023

**The wind farm pressure field**

Both Reviewers RC1 and RC2 provide careful and thoughtful comments on the manuscript. I will be able to incorporate their suggestion in the revised manuscript.

**Common criticisms of RC1 and RC2:**

Both reviewers RC1 and RC2 worry about the accuracy and utility of the Rigid Lid (RL) case that is described in my paper. They want to know how the RL case can be applied to the real world.  This is a fair criticism as I was mostly concerned with this RL case as a theoretical construct.  However, the RL case is needed to fully understand the origin of the pressure field.

In addition, both reviewers point out that the RL assumption is being used in some industry applications. Thus, my paper can also be used to understand and evaluate those applications. I will follow their advice and add such a discussion to the revised manuscript. This approach adds motivation to the RL analysis.

Both reviewers RC1 and RC2 request a more complete discussion of  the role of non-hydrostatic pressure fields near wind farms. I neglected these fields and did not fully explain why I did so. I agree with their criticism. I will add a complete explanation on non-hydrostatic dynamics in the revised manuscript.

Neither reviewer discussed the two methods ( Green's Function and Rankine Body) that I used to compute the RL pressure dipole and to illustrate the role of wind farm drag. This omission is slightly discouraging as these complementary methods underly one of the main insights in the paper.  I will try to clarify this point in the revisions.

Neither reviewer mentioned finding technical, logical or mathematical errors in the manuscript.

Both reviewers wish I had structured the paper differently by starting with a more complete description of the model. I will attempt to improve this in the revised version. Still, I think that both reviewers had a clear understanding of the formulation I used.

**RC1**

Main comments
1. At the end of the introduction I miss an outline of the paper. As no outline
is presented, it is currently not clear how the paper is structured. This
makes it harder to understand the relevance and significance of the paper.

**I will add a short outline to the manuscript.**

2. The comparison of the case with atmospheric gravity waves versus a rigid lid scenario is also interesting in light of the two main approaches for large-eddy simulations of wind farms that can be found in the literature, i.e., either resolving the atmospheric boundary layer and part of the free atmosphere (which supports the formation of gravity waves) or using a pressure-driven boundary layer with a rigid lid condition. A short literature survey and a quantitative discussion on how different the two approaches are would therefore be useful.

**I was not aware of previous RL treatments. I will add such a discussion.**

3. The description of the model in section 2 is too limited. Part of the governing equations are in fact shown in section 3, so why not present them when introducing the model? Furthermore, on line 109 it is stated that "the pressure field p(x,y) is derived using the hydrostatic assumption," but it is not clear to me what is meant by this.

**I will clarify how the hydrostatic assumption is used. I will be much more explicit concerning the role of non-hydrostatic pressure fields.**

4. Sections 3-10 all seem to focus on the rigid lid case, but the link back to the realistic case is missing a bit. What do the conclusions for the rigid lid case mean for the more realistic case? How different are the results when the inversion is not a perfect rigid lid?

**Table 1 already shows how the GW and RL cases differ quantitatively. I will expand the discussion of this Table.**

5. I have the impression that some of the theoretical results are in line with previous findings in the literature. For example, equation 25 shows that upstream flow blockage increases with farm width, and this has been found before (e.g. Allaerts & Meyers (2019). Same for the decay of the perturbation away from the farm (see eq 19 and 22), has this behavior been observed before (not sure myself)? It would be worthwhile to tie obtained results with what has been found before.

I will try, but an analysis of farm shape is not the subject of the paper.

**RC2**

 **Main comments**
1) *Reliability and significance of the rigid lid results*

RC2 makes two points here. First, that the RL model is not as accurate as the GW model. This is probably true, but no one has ever offered closed form expressions for the GW model.  The point of the paper was to show a qualitative agreement between GW and RL pressure fields and then to derive closed form expressions for the RL case. This approach pushes our understanding forward.

Second, RC2 points out that the industry uses RL models and thus my RL results can be used to evaluate them. I had missed this point and will add such a discussion.

*2) Wind-farm-induced pressure field without stratification*

I agree with RC2 that I should be much more explicit about the role of non-hydrostatic pressure contributions.

*3) More description*

Yes. I will add these additional five descriptions to the manuscript.

*4) Utility of measuring pressure in the field*

I think RC2 may overstate the accuracy of our farm drag estimates. For, example, the ocean skin friction associated with the farm-disturbed ocean wave field is substantial and is difficult to estimate. The pressure field may provide a useful check on the total drag.

---

## Author Response (AR1)

WES-2023-56 | Research article

Revision Date: August 16, 2023

**General responses to reviewers  RC1 and RC2:**

Both reviewers wish I had structured the paper differently by starting with a more complete description of the model.

**I revised the description of the model by moving the governing equations to the front. (Line 66-75)**

Both reviewers RC1 and RC2 worry about the accuracy and utility of the Rigid Lid (RL) case that is described in my paper. They want to know how the RL case can be applied to the real world.  This is a fair criticism as I was mostly concerned with this RL case as a theoretical construct.

**I have improved the theoretical discussion of the RL model and added a section on the application of the RL model to industrial models (Lines 245-258)**

Both reviewers RC1 and RC2 request a more complete discussion of  the role of non-hydrostatic pressure fields near wind farms.

**I have added a complete explanation of non-hydrostatic dynamics in the revised manuscript (Lines 85-89).  I also added a paragraph to the introduction about the cause of the pressure field (Lines 52-55).**

**Major Manuscript Changes**

1. **(lines  52-55) Added a short paragraph about the causes of the pressure field.**
2. **(lines 59-63) Added a brief outline of the paper at the end of the introduction.**
3. **(lines 66-75) Moved the governing equations forward to make the presentation more logical.**
4. **(lines 85-89) Added short paragraph about the non-hydrostatic part of the pressure field.**
5. **(lines 104-107) Added a better explanation of Table 2.**
6. **(lines  145-174) Interchanged the paragraphs on pressure cause and pressure impact to make the presentation more logical.**
7. **(lines  249-258) Added a short paragraph about the use of the RL assumption in industrial models.**
8. **(lines  278-299) Shortened and simplified the conclusions.**

**Specific Responses to RC1**

Main comments

1. At the end of the introduction I miss an outline of the paper. As no outline

is presented, it is currently not clear how the paper is structured. This makes it harder to understand the relevance and significance of the paper.

**I have added add a short outline to the manuscript. (Lines 59-63)**

2. The comparison of the case with atmospheric gravity waves versus a rigid lid scenario is also interesting in light of the two main approaches for large-eddy simulations of wind farms that can be found in the literature, i.e., either resolving the atmospheric boundary layer and part of the free atmosphere (which supports the formation of gravity waves) or using a pressure-driven boundary layer with a rigid lid condition. A short literature survey and a quantitative discussion on how different the two approaches are would therefore be useful.

**Thanks. I have added a brief new section on how my RL results could be used in "industrial" RL models.(Lines 249-258)**

3. The description of the model in section 2 is too limited. Part of the governing equations are in fact shown in section 3, so why not present them when introducing the model? Furthermore, on line 109 it is stated that "the pressure field p(x,y) is derived using the hydrostatic assumption," but it is not clear to me what is meant by this.

**Yes, I agree. I have moved the governing equation forward (Lines 66-75). I have also added discussion of the non-hydrostatic effects (Lines 85-89).**

4. Sections 3-10 all seem to focus on the rigid lid case, but the link back to the realistic case is missing a bit. What do the conclusions for the rigid lid case mean for the more realistic case? How different are the results when the inversion is not a perfect rigid lid?

**Table 2 already shows how the GW and RL cases differ quantitatively. I expanded the discussion of this Table.**

5. I have the impression that some of the theoretical results are in line with previous findings in the literature. For example, equation 25 shows that upstream flow blockage increases with farm width, and this has been found before (e.g. Allaerts & Meyers (2019). Same for the decay of the perturbation

away from the farm (see eq 19 and 22), has this behavior been observed before (not sure myself)? It would be worthwhile to tie obtained results with what has been found before.

**I take your point, but an analysis of farm shape is not the subject of the paper. I would rather not get into this.**

Specific comments
1. Line 19: "... the wind slowing by these farms ... . This issue has an extensive literature, ..." I find this statement about the wind slowing a bit vague. Do you refer to the upstream flow deceleration, the wind deficit behind individual turbines, slowing down of the atmospheric boundary layer above the wind farm? All of these? Please make this more clear.
**This comment is in the space-limited abstract. There is no room there for more detail in the abstract.**

2. Line 25: "In a stably stratified atmosphere, ..." I think this statement can be misleading for the reader. Gravity waves only form in regions where the flow is stably stratified, but you can get gravity waves also above neutral and unstable boundary layers. The statement could be misinterpreted as if gravity waves only occur when the atmospheric boundary layer is stably stratified (i.e. when the surface heat flux is negative).
**I have improved the description of the model formulation..**

3. Line 28: I believe the term used by Bleeg (and many others) is blockage rather than blocking.
**OK. I changed it to Blockage.**

4. Line 29: "... over the farm, it can fight back against the turbine drag, ... . Finally, it alters the recovery of the wake." Have these effects been observed in the literature? Please cite relevant studies.
**This was discussed in my first paper (Smith 2010). This paper is cited.**

5. Line 33: "According to Gribben and Hawkes (2019), the local non-hydrostatic pressure disturbances decay inversely ..." Please clarify whether this result is for a single turbine or for a farm.

**I improved my discussion of non-hydrostic effects and the Gribben/Hawkes paper (Lines 85-92)**

6. Line 34: "The farm-generated hydrostatic pressure disturbance is more far-reaching." What do you base this statement on? Evidence from literature (if so, add references), or is this based on your own findings (then say something like "as will be shown in this study, …").

**OK. I modified this language.**

7. Line 43: Is there a specific reason why the model is limited to hydrostatic gravity waves?

**The linearized model could be extended to non-hydrostatic cases, but those effects are small for farms larger than a few kilometers. I have added a discussion of this point. (Lines 85-92)**

8. Line 53: Specify what you mean by "wrapping"

**OK. I added a mention of the FFT method giving periodic wrapping.**

9. Line 54: Please define "DAR" more clearly (I assume rotor disk area to covered surface area, where the latter is assumed to be $s_x s_y D^2$).

**OK. I rewrote this sentence to be clear (Line 98).**

10. Line 86: "When N=0, the displacement approaches zero as $1/g'$ and when $g'=0$ it approaches zero as $1/N$." This relationship would be more clear when plotted in a figure.

**I prefer the table as I could quantify more variables, each with different units.**

11. Improve caption of table 2. Currently, it is not clear what is listed and how some of the parameters are defined (e.g. for the definition of Gamma one needs to refer to the main text).

**OK. I improved the caption for Table 2 and the text. (Line 104)**

12. Line 109: "When this equation is solved for the perturbation wind, the scalar wind deficit is computed from ..." This transition is too fast, I needed to look up the definition in Smith (2022) to understand that the definition results from a linearization. Please explain more clearly how the scalar wind deficit is obtained.

**I already cited a reference for this. I now added a reminder about the linearization.(Line 74)**

13. Line 116: "Because the pressure field p(x,y) decays at infinity, it does not influence TD ..." Again, going a bit fast. The pressure term vanishes because of the divergence theorem and the fact that p decays at infinity, but for this you also need the continuity equation to go from U $\cdot$ $\nabla$p to $\nabla$ $\cdot$ Up.

**This proof is just the Fundamental Theorem of Calculus because U is a constant. When you integrate dp/dx you get the difference in p at infinity. This difference is zero if the disturbance decays.**

14. Line 122: "In 2-D non-divergent flow ..." The assumption of non-divergent flow can be made because of the rigid lid case, or does this also hold in the GW case? Please clarify.

**OK. I clarified the text to say that the rigid is required.**

15. Figure 3: Why do you apply a low pass filter to (b)? What high-frequency content are you filtering out? Noise due to the low resolution?

**The Laplacian involved two derivatives that amplify noise. The RL case has sharp gradients because the farm has sharp boundaries.**

16. Line 146: "The linearized Bernouilli equation ... is approximately valid upwind, ..." Upon what is this statement based? Did you check whether this approximation is valid?

**This result follows from the governing equations (1) at F=0 upwind and the Rayleigh friction is small.**

17. Line 156: "The magnitude of this ratio increases with aspect ratio …" Not clear to me why this would be the case. Did you try other aspect ratios?

**I did a few other aspect ratios. I did not want the reader to over generalize. This paper was not intended to discuss farm shape.**

18. Line 170: "… the wake decay length (L=U/C) …" Is this wake decay length defined in literature? If not, it should be made clear why the wake decay length is defined like this.

**I dropped this comment to simplify the sentence. It is true however that the wake decay length is U/C.**

19. Line 174: "In incompressible or non-divergent flow, …" In the GW case, the flow is also incompressible, but flow divergence is allowed and leads to inversion displacement. Does that mean that the role of the pressure as you describe only holds for incompressible and non-divergent flow?

**No. In the rigid lid case the flow is horizontally non-divergent so the pressure is determined diagnostically. I had added some text on this point.(Lines 52-55)**

20. Line 281: "The stability values need to be an order of magnitude larger before the rigid lid approximation becomes quantitatively accurate." What criteria is this statement based on? When is the rigid lid approximation considered quantitatively accurate? This needs more explanation.

**The reader can judge from Table 2.**

21. Line 296: "… including the change in ocean surface wind stress caused by turbine induced boundary layer turbulence." Not very clear, and also not sure why this is relevant (not discussed anywhere else in the paper).

**The drag from a wind farm comes about 10% from the ocean skin drag associated with the rough ocean surface. This leads to uncertainties. That was my point.**

Technical comments

1.  Line 28: so call "Blocking" → so called "Blockage" (see also earlier comment)

**OK. I used "blockage". In mountain meteorology, the same upstream slowing is called "blocking".**

2.  Table 2, entries for parameter A for cases g'=0.05, 0.1, and 0.2 with N=0: The value for A is exactly equal to zero. Is this correct or is this a typo?

**Good question. These are supercritical cases and there is no dipole pressure field. I changed the zeros to N/A.**

3.  Section 4 is missing?

**Fixed**.

**Specific Responses to RC2**

**Main comments**

1) *Reliability and significance of the rigid lid results*

**RC2 makes two points here. First, that the RL model is not as accurate as the GW model. This is probably true, but no one has ever offered closed form expressions for the GW model. The point of the paper was to show a qualitative agreement between GW and RL pressure fields and then to derive closed form expressions for the RL case. This approach pushes our understanding forward.**

**Second, RC2 points out that the industry uses RL models and thus my RL results can be used to evaluate them. I have added such a discussion. (Lines 249-258)**

*2) Wind-farm-induced pressure field without stratification*

I **agree with RC2 that I should be much more explicit about the role of non-hydrostatic pressure contributions. I have added this discussion.**

*3) More description*

**Yes. I added these additional five descriptions to the manuscript.**

*4) Utility of measuring pressure in the field*

**I think RC2 may overstate the accuracy of our farm drag estimates. For, example, the ocean skin friction associated with the farm-disturbed ocean wave field is substantial and is difficult to estimate. The pressure field may provide a useful check on the total drag.**

**Minor comments**

Line 28: 'so call "Blocking"'. "Blockage" is the more commonly used term in the wind energy community. It is also the term used in the paper. "so-called" should probably modify "blockage" rather than "so call".

**OK. I have called it "blockage". The literature on mountain meteorology calls it "blocking".**

Line 80: "the pressure field increased from zero." I think this phrasing can be improved upon. I'm pretty sure I understand what the author means when he writes that the "pressure field increased", but maybe the "pressure disturbances increased" would be better.

**Yes. I have reworded this part.**

Line 122: It may be worth pointing out that the right-most term in the equation is also zero.

**Yes. I rewrote this section**.

Lines 172 and 176: The sentence starting "The Rayleigh force in this case is non-divergent…" on line 176 appears to repeat a point just stated on line 172 in the sentence starting "This independence of the pressure field…" Or perhaps I've missed a subtlety. If not, perhaps the second sentence should be deleted.
 **I have rewritten this section**.

Line 294: There is a typo at the end of the line between "is" and "proportional".

**Fixed**

---

## Referee Report (RR1)

The revised manuscript addresses nearly all the comments in my original review. I have concerns about some of the statements in the paper related to using the rigid lid assumption in industrial/engineering wind farm flow models, but these are not central to the paper and do not require changes. I support publication after some very minor fixes.

**Main comment:** As requested, the author now includes a discussion on the use of the rigid lid (RL) assumption in industrial/engineering wind farm flow models. I infer from the text that the author and I have different views on the soundness of this modeling choice. It is possible that these differences may amount to little more than a disagreement on emphasis. Nevertheless, at the risk of belaboring a point made in my initial review, I would like to write a few more words on the matter.

In the absence of a gravity wave formulation, I would agree that including the rigid lid in an engineering model, such as that from Gribben and Hawkes, is probably better than not having it. However, the results of this paper and my own experience indicates that there is high risk that this modeling assumption will yield large errors. Figures 3 and 4, which compare RL and gravity wave (GW) model results, suggests that the predicted energy yield would frequently be very different between the two model types. The author makes a similar point in the Discussion section, but also, separately, in a response to my initial review, he points out that while GW is more likely to better represent reality, we cannot be sure of it. Fair point. Nevertheless, I believe the large quantitative differences in the results from the two models suggest an enhanced risk of large errors in models using RL, and this risk should be investigated, for example, by using numerical experiments executed with higher fidelity models. Although this last comment is not directly relevant to the paper under review--and does not necessitate more edits--the importance of this subject is large enough within the wind energy industry that I felt I needed to put this point in writing.

**Minor comments:**

Line 22, third line of the abstract: The word "modify" has been replaced with "slow". There are two issues here. Firstly, the change was not redlined. The main changes to the manuscript have been redlined, but many smaller changes have not. It would have been easier to review this revision if more of the changes had been redlined. Secondly, I think the original wording ("modify the wake recovery") is better. The pressure gradients may "slow the wake recovery" downstream of the wind farm, but not within the wind farm, where these pressure gradients can speed the wake recover.

Line 98: "farm area" is a bit ambiguous. I suggest adding "planform" before "farm".

Lines 52 and 53: Fig 3b should be Fig 4b.

---

## Author Response (AR2)

Response to reviewer and editor for WES-2023-56

**The wind farm pressure field**

(October 22, 2023)

The new comments mention five points. Responses are below.

1. No Plot

I understand your suggestion about the plot from Table 1, but I had thought about this choice before I submitted the paper and had good reasons for not doing that. First there are five dependent variables in the Table; four that vary weakly (max deficit, Gamma, delta P and dipole strength (A)) and one (max displacement) that varies by many orders of magnitude. The first four would each require their own plot scale. The max displacement variable follows a simple power law line that shows nicely on a log-log plot but is also neatly summarized by one sentence in the paper (line 118-119). This sentence is much more compact that a plot. The advantage of the Table is that it provides precise model-derived values (to three significant figures) that other researchers can use to check my work. Plots cannot be read that precisely.

2. Industrial RL models (Line 256)

I was happy to follow the reviewer's suggestion to add the Industrial RL paragraph, but I am certainly not promoting such models. If the reader has mis-interpreted my intent, it may be due my poor wording in Line 256. I have rewritten this sentence to make it clearer.

3. Blocking vs Blockage (line 44)

OK. I added a comment about this terminology in Line 44.

4. Wake recovery (line 22)

I added the word "farm" in line 22. I went back to the generic term "modify", even though the pressure effect is mostly to slow the wake recovery.

5. Wrong figure number (line 152-153)

I fixed the wrong figure number in line 152-153. Thanks for catching that.

---

## Author Response (AR3)

WES-2023-56

December 17, 2023

Note to Editor:

Thanks for accepting this paper. In handling this final version, please note

1. All the equations are done in Microsoft Word. Thus, the units appear in italics.
2. The abbreviation GW for gravity waves is defined several times in the text.  GW is never used in this paper to mean GigaWatts.
3. I corrected the symbols for Newtons (N) and Pascals (Pa)
4. I slightly shortened Table 2 to make it more readable.
5. There are four figures in the paper.  Each figure has two parts. I have submitted the figures as png files, combined into a ZIP file.
6. The reference list format is in pretty good shape but two references (Hansen 1983 and Gribben 2019) are non-standard.

I hope you can repair any mistakes in formatting.

Thanks

Ron Smith